# Peer and lay health work for people experiencing homelessness: A scoping review

**Jessica Mangan**[1]\*, **Pablo del Cid Nunez**[2], **Sara Daou**[3], **Graziella El-Khechen Richandi**[3], **Amna Siddiqui**[3], **Jonathan Wong**[4,5], **Liz Birk-Urovitz**[3], **Andrew Bond**[6,7,8,9], **Aaron M. Orkin**[2,6,8,10]

1 School of Public Health and Social Policy, University of Victoria, Victoria, British Columbia, Canada, 2 Dalla Lana School of Public Health, University of Toronto, Toronto, Ontario, Canada, 3 Population Health Services, Inner City Health Associates, Toronto, Ontario, Canada, 4 Family Health Team, Inner City Health Associates, Toronto, Ontario, Canada, 5 St Michael's Hospital, Toronto, Ontario, Canada, 6 Inner City Health Associates, Toronto, Ontario, Canada, 7 National Health Fellow, McMaster University, Hamilton, Ontario, Canada, 8 Department of Family and Community Medicine, University of Toronto, Toronto, Ontario, Canada, 9 Canadian Network for the Health and Housing of People Experiencing Homelessness, Toronto, Ontario, Canada, 10 MAP Centre for Urban Health Solutions, Unity Health, Toronto, Ontario, Canada

\* jessica.mangan@icha-toronto.ca

**Data Availability Statement:** All relevant data are within the paper.

**Funding:** The authors received no specific funding for this work.

## Abstract

Homelessness poses complex health obstacles for individuals and communities. Peer and lay health worker programs aim to increase access to health care and improve health outcomes for PEH by building trust and empowering community-based workers. The scope and breadth of peer and lay health worker programs among PEH has not been synthesized. The primary objective of this scoping review is to understand the context (setting, community, condition or disease) encompassing peer and lay health worker programs within the homelessness sector. The secondary objective is to examine the factors that either facilitate or hinder the effectiveness of peer and lay health worker programs when applied to people experiencing homelessness (PEH). We searched CINHAL, Cochrane, Web of Science Core Collection, PsycINFO, Google Scholar and MEDLINE. We conducted independent and duplicate screening of titles and abstracts, and extracted information from eligible studies including study and intervention characteristics, peer personnel characteristics, outcome measures, and the inhibitors and enablers of effective programs. We discuss how peer and lay health work programs have successfully been implemented in various contexts including substance use, chronic disease management, harm reduction, and mental health among people experiencing homelessness. These programs reported four themes of enablers (shared experiences, trust and rapport, strong knowledge base, and flexibility of role) and five themes of barriers and inhibitors (lack of support and clear scope of role, poor attendance, precarious work and high turnover, safety, and mental well-being and relational boundaries). Organizations seeking to implement these interventions should anticipate and plan around the enablers and barriers to promote program success. Further investigation is needed to understand how peer and lay health work programs are implemented, the mechanisms and processes that drive effective peer and lay health work among PEH, and to establish best practices for these programs.

**Competing interests:** The authors have declared that no competing interests exist.

# Background

Homelessness describes the situation of an individual without stable, safe, permanent, appropriate housing, or the means and ability of acquiring it" [1]. Homelessness converges with various intersecting identities and forms of marginalization, particularly affecting Indigenous and refugee populations and specific definitions of homelessness have been developed to encompass these diverse considerations [1]. Based on current estimates, more than 235,000 people in Canada experience homelessness in any given year, and 25,000 to 35,000 people may be experiencing homelessness on any given night [2, 3]. In the United States, over 580,000 people may be experiencing homelessness on a single night [4]. Homelessness is a growing concern in many countries and can lead to a range of negative health, social and economic outcomes for those affected [5]. For example, treatment and prevention of health issues may be neglected due to various barriers such as competing needs for food and shelter, lack of access, stigmatization and discrimination, financial barriers, complex documentation requirements, and limited health literacy [6].

However, the scope and operational enablers and inhibitors of peer and lay health programming among people experiencing homelessness has not been synthesized. We conducted a scoping review to identify evidence relating to peer and lay health worker programs related to people experiencing homelessness to further identify the key characteristics and inhibitors/enablers of these programs.

# Methods

## Definition of peer and lay workers

Peer and lay support workers, who are individuals without professional training but with a commitment to helping others, can provide support to people experiencing homelessness by offering their expertise, practical assistance and a sense of community. "Lay" refers to a community member who has received some training to promote health or to carry out some healthcare service, but is not a health care professional [7]. "Peer" refers to a community member who shares similar life experiences to the community with which they work [8]. Peer support workers can draw on their lived and living experience to provide support to others experiencing similar situations. Peer support work can be defined by offering and receiving help, based on shared understanding, respect and mutual empowerment between individuals in similar scenarios [8]. The use of peer support has been long established in the mental health sector and has recently been implemented in other service areas such as substance use treatment, harm reduction, chronic disease management, homelessness, and sex work [9–11]. Various terms for lay and peer support worker are used interchangeably in the literature such as, peer health worker, peer specialist, peer advisor, lay support worker, lay health worker, community health worker, community support worker, peer ambassador, health ambassador, among others. [7–11].

## Question and objectives

This scoping review aimed to map the extent, range, and nature of literature on the engagement of lay and peer workers in health services for people experiencing homelessness. The participants in this review were people experiencing homelessness (PEH). The concept explored was characteristics of effective peer and lay health worker programs in the homeless sector. In terms of context, the review considered studies that included people experiencing homelessness participating in any type of peer or lay health worker programs. The primary objective of this scoping review was to understand the context (setting, community, condition or disease)

encompassing peer and lay health worker programs within the homeless sector. The secondary objective was to examine the factors that either facilitate or hinder the effectiveness of these programs when applied to PEH.

## Protocol and registration

The scoping review was conducted based on the JBI methodology for scoping reviews [12] and reported according to the Preferred Reporting Items for Systematic Reviews and Meta-Analyses extension for Scoping Reviews (PRISMA-ScR) [13]. A scoping review protocol was developed and registered with the Open Science Framework (osf.io/u4yp8).

## Information sources

A preliminary search of Medline (Ovid), the Cochrane Database of Systematic Reviews and *JBI Evidence Synthesis* was conducted. A review by Barker and Maguire in 2017 [14] reported on the effectiveness of intentional peer support on young adults and adult homeless persons. That review found 11 articles describing ten studies that examined the focus of intentional peer support with a homeless population; demonstrating limited evidence of intentional peer support with a homeless population [14]. This review was conducted prior to the COVID-19 pandemic, and substantial new work in the field has been published in the intervening years [15, 16]. A second review by Lloyd-Evans and colleagues in 2014 [17] reported the impact of peer support for people with severe mental illness, but did not specifically address the population of people experiencing homelessness. This initial search was validated using a set of ten eligible papers [14, 18–25] to ensure that papers with one or more of the relevant themes were retrieved by our search strategy.

We conducted an electronic search of MEDLINE, CINHAL, Cochrane, Web of Science Core Collection, PsycINFO, and Google Scholar to identify peer-reviewed publications. The search strategy was drafted by the lead author and refined in consultation with peer-reviewers. The final search strategy for MEDLINE can be found in S1 Text. The search strategy, including all identified keywords and index terms, was adapted for each included database and/or information source. The reference list of all articles was screened for additional studies. The listed databases were searched up until 21 February 2023. Eligible studies were limited to the English language.

## Selection of sources of evidence

Eligible papers included studies of any kind that described peer or lay health workers engaged in any kind of health or social program serving people experiencing homelessness. Our review adopted Gaetz and colleagues' definitions of homelessness referring to an individual without stable, safe, permanent, appropriate housing, or the means and ability of acquiring it [1]. This includes

1. Unsheltered, including those living on the streets or in places not intended for human habitation;

2. Emergency sheltered, including those staying in overnight shelters for people who are homeless and;

3. Provisionally accommodated, referring to those whose accommodation is temporary or lacks security [1].

Literature referring to other more inclusive definitions of homelessness, such as the Indigenous Definition of Homelessness in Canada, were also included in the review. Specifically,

Indigenous homelessness is not defined as lacking a structure of habitation, rather it is described and understood through a composite lens of Indigenous worldviews [1]. Participants were not limited to a specific age. Studies that included adults, youth and children of all ages that were experiencing homelessness were considered.

The concept explored in this scoping review was the characteristics of effective peer and lay health worker programs in the homeless sector. Studies applying the use of peer and lay health worker programs within the homeless sector were included in this review. We used the term "peer and lay health worker" programs to refer to a wide range of care redistribution strategies that involve the deliberate integration of non-health professionals into the health workforce team, and especially people with lived experience of homelessness or lived experience with the health condition targeted by the given program. A wide range of terms are used to describe these programs, including peer workers, lay workers, community health workers, health volunteers, and health ambassadors, among others [14, 20]. Eligible studies were not limited by geographic location.

This scoping review considered analytical observational studies including cohort studies, case-control studies and cross-sectional studies. This review also considered descriptive observational study designs including case series, individual case reports and descriptive cross-sectional studies for inclusion. Qualitative studies were considered that focused on data including, but not limited to, designs such as phenomenology, grounded theory, ethnography, qualitative description and action research.

Following deduplication, citations were uploaded to *Covidence* software [26] and screened by two independent reviewers for assessment against the inclusion criteria for review. The full text of selected citations was assessed in detail against the inclusion. Reasons for exclusion of sources of evidence at full text that did not meet the inclusion criteria were recorded and reported. All disagreements on document selection were resolved through consensus.

## Data extraction and synthesis

We developed an online data extraction tool using Microsoft Forms and Microsoft Excel. The data extraction tool provided with the protocol registration was adapted following initial data abstraction efforts to better reflect the nature of the available data and the objectives of this scoping review. Data was extracted by the lead author. Extracted data included details about the participants, concept, context, study methods and key findings relevant to the review questions. For example, the disease/condition; outcome measured; characteristics of peer workers (i.e., age, gender, background); and training provided to the peer worker.

Context of peer and lay health worker programs, enablers and inhibitors/barriers reported in the included documents were recorded. The lead author extracted text verbatim and synthesized the enablers and inhibitors into themes. We then reported on study characteristics and these themes through narratives and tables.

## Results

### Selection of sources of evidence

A total of 4119 records were identified in CINHAL, Cochrane, Web of Science Core Collection, PsycINFO, Google Scholar, and MEDLINE. Following the removal of duplicates, 2536 records were identified as potentially relevant for this study. Following title and abstract review, 227 records were considered relevant for full-text review. Based on the inclusion and exclusion criteria, 38 documents were included in the scoping review (See Fig 1).

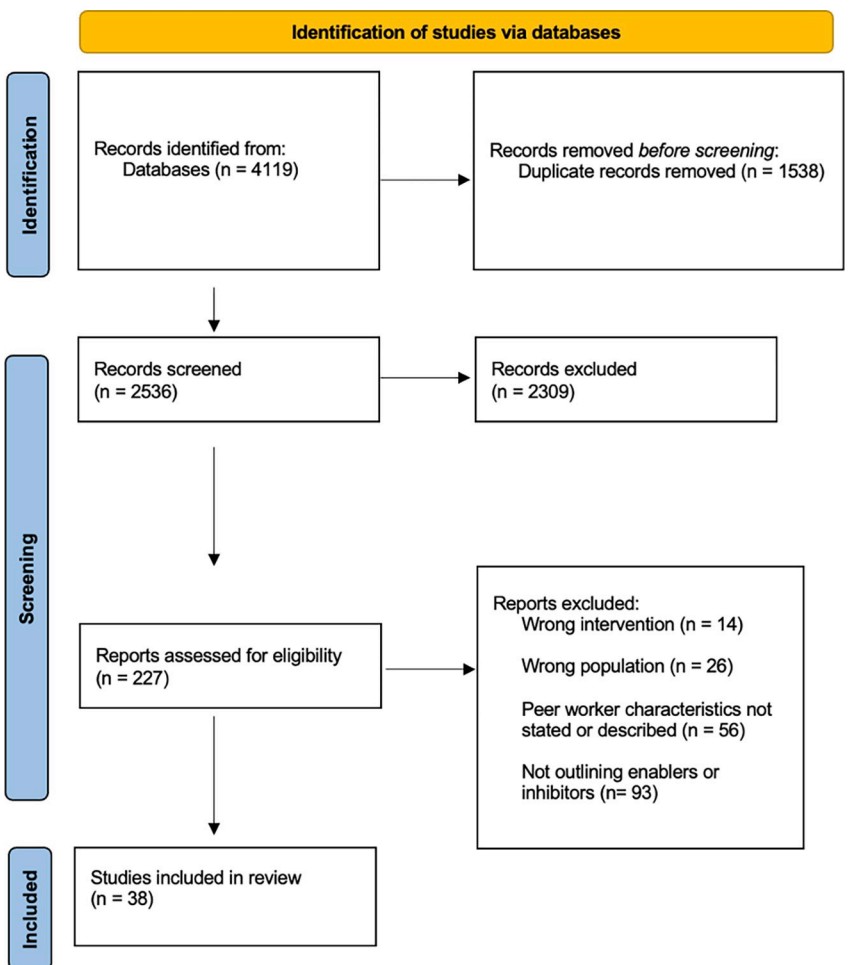

**Fig 1. PRISMA flow diagram showing inclusion and exclusion strategy.**

## Characteristics

38 documents were included in the scoping review. Studies included systematic reviews, qualitative studies, pilot studies, case studies, community-based participatory research, program evaluation, randomized control trials and mixed-method studies. The earliest document was published in 1995, with the remaining being published from 2005–2023. 20 sources described programs in the United States [16, 19, 20, 22, 27–42], four sources described programs in Canada [43–46], nine sources described programs in the United Kingdom [14, 18, 21, 47–52], and there was one source from each in Kenya [53], the Netherlands [54], and Australia [55]. Two knowledge syntheses were included that referred to studies from various locations [56, 57]. See Table 1 for an overview of the included documents.

## Populations and personnel

All included studies focused on interventions for people experiencing homelessness combined with a specific socio-demographic or clinical need, such as Veterans experiencing homelessness, COVID-19, mental illness, drug or alcohol use, tuberculosis, diabetes, HIV, Hepatitis C, and intimate partner violence [16, 20, 27–30, 43, 47, 48].

**Table 1. Overview of included documents.**

| Authors Location | Study Type | Condition | Interventions | Outcome reported | Peer worker characteristics | Training provided | Employment status |
|---|---|---|---|---|---|---|---|
| Barker & Maguire. (2017) United Kingdom | Scoping Review | Homelessness | Various; all studies had peers as part of their intervention. | Various (i.e., health, social, addiction outcomes). | Homeless, adult/ over 18; "shared experience and the ability to empathize and develop a mutual trust and understanding". | Various | Not reported |
| Barker et al. (2018) United Kingdom | Qualitative study | Homelessness | Utilizing peer support to overcome barriers | Social support, role modelling | 18 years and older, had experience with homelessness | Receive training on roles, boundaries, confidentiality, safeguarding, and pertinent homelessness issues. | Volunteers |
| Barker et al. (2019) Various countries | Qualitative study | Homelessness | Utilization of peer support in homelessness services. | The important elements of effective peer support for those experiencing homelessness. | Had shared experiences of phenomena, such as mental health, addiction and homelessness. | Professional organization training on peer support | Not reported |
| Blonigen et al. (2022) United States | Pilot study, mixed-methods | Homeless veterans | Weekly health coaching sessions with a peer over 12 weeks. | Session attendance and satisfaction; barriers/facilitators to implementation; and pre/post utilization of services. | Lived experience of homelessness and were in recovery from substance use and/or mental health problems. | 6-day virtual training. | Not reported |
| Cerna et al. (2023) Glasgow | Case study | Chronic homelessness | Utilization of peer navigators for healthcare | Status of tenancy, days in homelessness, and engagement of services. | Minimum of two-year abstinence from substance use; lived experience of homelessness and/or substance use. | Completed one year of addiction training. | Employed at regular, full-time, paid job positions. |
| Choi et al. (2022) Los Angeles, US | Qualitative | COVID-19 | Conduct outreach and assessing interest in a COVID-19 vaccine. | Uptake of COVID-19 vaccinations. | Person with lived experience of homelessness and has completed COVID-19 vaccination series. | Not reported | Employees |
| Corrigan et al. (2015) Chicago, US | Community-based participatory research–qualitative | Homelessness and mental illness | | To determine the potential of peer navigators for addressing the health needs of homeless African Americans with mental illness. | Homeless African Americans with mental illness | Not specified | Volunteers |
| Crisanti et al. (2017) New Mexico, United States | Longitudinal analysis | Mental health and addiction | Peer-delivered permanent supportive housing | 61% of all participants received supportive housing. Supportive housing was significantly associated with good to excellent health 6 months after baseline (odds ratio = 3.11, 95% confidence interval [1.12, 8.66]). | History of homelessness, mental illness and/ or addiction. | 40hrs of classroom training, certification exam, additional training in housing and supportive services delivery. | Employees |

*(Continued)*

**Table 1.** (Continued)

| Authors Location | Study Type | Condition | Interventions | Outcome reported | Peer worker characteristics | Training provided | Employment status |
|---|---|---|---|---|---|---|---|
| Croft et al. (2013) London, United Kingdom | Qualitative | Tuberculosis (TB), homelessness, and drug/ alcohol dependency. | Peers support clients accessing welfare, understanding tuberculosis and navigating the health care system. | Understanding the motivation of becoming a peer educator and the impact this has on the individual; to gain insight into the use of peers to support TB control. | Must have had treatment for active TB, experience of homelessness and/or drug/alcohol dependency, and have been a peer educator within the last 3 years of the project. | Not reported | Volunteers |
| David et al. (2015) Connecticut, US | Program evaluation | Homelessness and were diagnosed with substance use or comorbid substance use and mental illness. | Assisting clients with finding transportation; making/attending appointments; filling out housing paperwork; and reviewing job interviewing skills. | Demonstrated high rates of retention in program services; significant reductions in days of drug use, depression, anxiety, and hallucinations; higher rates of employment, etc. | Female and experience of substance use, mental illness and/ or homelessness. | Received training and supervision in providing peer support and culturally responsive, gender-specific interactions. | Employees |
| Davis, et al. (2016) Michigan, US | Implementation research | Diabetes (T2DM) | A 4-week peer-led diabetes education program | Increased knowledge on signs/symptoms, and complications of diabetes and diabetes medications. | History of homelessness, the use of Heartside Neighbourhood resources, history of T2DM, and taking medications for diabetes. | Peer-leader orientation was provided by a registered nurse each week for 1 month prior to sessions. | Not reported |
| Deering et al. (2009) Vancouver, Canada | Pilot study | Street-entrenched HIV-positive women | Peer-driven intervention, weekly peer support meetings, a health advocate system, peer outreach service. | Reported an increase in adherence throughout the program as well as improved viral load outcomes. | HIV+ women | Standardized training was led by a local sex work organization | Employees |
| Erangey et al. (2020a) Colorado, US | Participatory action research—qualitative | Young people experiencing homelessness | Support youth with goal setting, problem solving, skill-building, navigating healthcare systems, and building connections. | Peers center self-directed growth, invite possibility and create containers of hope. | Background of homelessness | Not reported | Employees |
| Erangey et al. (2020b) Colorado, US | Qualitative participatory design | Youth experiencing homelessness | Includes assisting clients in accessing the resources and developing the skills they need related to housing, healthcare, etc. | To determine the perspective of peer support specialists on rapport building, and processes related to youth experiencing homelessness. | Having experienced housing insecurity or street-based homelessness | Not reported | Employees |
| Flike et al. (2020) Boston, MA | Observational study | Women over 55 experiencing or at risk for homelessness | Implemented Bridges to Elders (BTE) program | Improved housing status, primary care access, and targeted health outcomes for older women who are experiencing or at risk for homelessness. | Not reported | Not reported | Employees |

(*Continued*)

**Table 1.** (Continued)

| Authors Location | Study Type | Condition | Interventions | Outcome reported | Peer worker characteristics | Training provided | Employment status |
|---|---|---|---|---|---|---|---|
| Fors & Jarvis. (1995) United States | Quasi-experimental; non-random | Drug abuse | Four-session program on drug use presented by peer youth leaders. | Peer-led groups had significant difference in knowledge gained; clients were more willing to accept responsibility for their actions; increased willingness of the clients to contact helping resources for their friends. | Not reported | Three-day training in the curriculum; provided with a detailed resource manual. | Employees |
| Herts et al. (2020) Massachusetts, US | Qualitative descriptive study for the purpose of program evaluation | Homelessness | Community health workers (CHW) perform five key roles: relationship-building, social support, system navigation for housing, system navigation for health care, and community engagement. | Two outcomes: (1) social support is a crucial component of the CHW role; and (2) close supervision is essential in order to balance CHW flexibility and consistency. | Because of the small number of CHWs, their characteristics are not displayed in order to protect anonymity. | Not reported | Employees |
| Kidd et al. (2019) Toronto, Ontario | Mixed methods | Youth experiencing homelessness | Outreach to supportive housing setters/shelters; 1–1 peer support; facilitation of social outings; co-participation in the mental health group; and participatory action projects. | Youth who were more engaged in peer programming made more gains in key life areas. | Had a lived experience of homelessness, were at the time experiencing a stable situation with respect to housing and other life domains, and had demonstrated excellent engagement and leadership skills. | Peers completed mandatory trainings with respect to organization expectations and ethics. | Employees |
| MacLellan et al., 2017 London | Qualitative | Hepatitis C in people who have a history of injecting drug use and homelessness | The peer ambassador (PA) role was to engage with referred clients, support and advocate for them through the appointment process and within peripheral services. | Strategies that PAs use to achieve connectedness through establishment of a positive therapeutic relationship with clients. | Experience of recent homelessness, substance misuse and mental health challenges; personal experience of hepatitis C; male; average age of 48. | Not reported | Not reported |
| Magwood et al., 2019 Canada, USA, UK and Australia | Systematic review | Homelessness | Various peer support interventions | Behavioural and structural factors that influence the acceptability of health and structural interventions. | Various depending on each program (i.e., opioid users; women only). | Various | Not reported |

(*Continued*)

**Table 1.** (Continued)

| Authors Location | Study Type | Condition | Interventions | Outcome reported | Peer worker characteristics | Training provided | Employment status |
|---|---|---|---|---|---|---|---|
| Miler et al., 2020 Various countries | Systematic review | Homelessness | Various | Address presenting research on peer support for substance use and housing challenges transparently and explore embedding interventions without adding stress to participants. | Various | Various | Various |
| Moledina et al., 2021 United States | Systematic review | Homelessness | Various peer support services for PEH | Peer support programs demonstrated no impact on housing relative to usual care; no economic evidence found for peer support. | Employs workers with shared life experiences to provide social support, advocacy, education, and role modelling to homeless individuals. | Not reported | Employees |
| Morris et al., 2020 West Midlands, UK | Intervention study | People who inject drugs (PWID)/ Hepatitis C | Point of care testing and treatment for PWID | Project was received well by homeless PWID, hepatology treatment teams, hostel and homeless services, reflecting on the constructive impact on collaborative working. | Not stated | Worked alongside a clinical nurse specialist | Not reported |
| Nyamathi et al., 2021a Los Angeles US | Clinical trial | Latent TB | RN/CHW-based program across the LTBI continuum of care that delivers 3HP treatment for homeless adults; CHWs provided weekly one-on-one, case management sessions. | The RN/CHW program achieved a 91.8% 3HP treatment completion rate among homeless adults. | Not stated | Trained to deliver the 3HP LTBI intervention. | Employees |
| Nyamathi et al., 2021b Los Angeles, US | Qualitative community-based participatory design | Hepatitis C | Making referrals to medical and social services and housing; bring one dose of the medication to the client each day with education | Essentials of successful program design, participant engagement and retention | Provided educational and social services for homeless clients, and worked in a homeless clinic for at least a year | Not reported | Employees |
| Parkes et al., 2022 Scotland and England | Mixed-methods | People experiencing homelessness and problem substance use | Developed relationships with participants; they worked with (and often accompanied) them to access services, such as substance use treatment, health care, housing and welfare/benefits. | There was reduced drug use and an increase in the number of prescriptions for opioid substitution therapy; there were reductions in risky injecting practice and sexual behaviour. | Someone with lived experience of homelessness and substance use | They received training on a range of topics, including harm reduction, trauma and naloxone administration. | Employees |

*(Continued)*

**Table 1.** (Continued)

| Authors Location | Study Type | Condition | Interventions | Outcome reported | Peer worker characteristics | Training provided | Employment status |
|---|---|---|---|---|---|---|---|
| Ponce et al., 2014 United States | Participatory, action-based qualitative and quantitative evaluation | Women experiencing intimate partner violence | Conducted outreach and drew on their lived experiences to engage program participants | Identified recommendations (i.e., non-judgmental approach, offering multiple meeting sites, training in trauma, and utilizing peer engagement). | Came from abusive families, were in recovery, know and grew up with some of the clients, were mothers, and were in relationships with men or women that sometimes bore similarities to those of the clients | Received training on gender, trauma, and culture from a statewide training and advocacy organization that focuses on women's issues in behavioral health | Employees |
| Resnik et al., 2017 United States | Randomized control trial | Homeless Veterans | Mentors routinely followed-up with each of their assigned Veterans per the care plans identified in the clinic visit. Veterans were followed-up for a total of 6 months. | 65% of the subsample and 83% of the full sample benefited from a peer mentor. Participants who benefited had more peer visits and minutes of intervention, were more likely to be a minority, and were less likely to have posttraumatic stress disorder. | Formerly homeless Veterans | Peer mentors underwent extensive training (includes case-management and peer mentor team intervention for homeless Veterans with co-occurring mental illness and substance use). | Not reported |
| Rosen et al., 2023 Los Angeles, California, US | Program Evaluation | COVID-19 | Mobile vaccination program | COVID-19 vaccination uptake for PEH; 16% of participants cited their conversation with staff/peer workers as a primary reason for deciding to be vaccinated. | Not reported | Motivational interviewing, trauma-informed and vaccine education | Employees |
| Salem at al., 2020 Los Angeles, US | Qualitative study | Tuberculosis | Weekly 20-min case management session; assess medication side effects, track missed doses, provide health related support | A nurse-CHW partnered intervention delivery method improved 3HP LTBI recruitment, delivery, retention and completion among homeless adults at high risk for active TB | Not reported | Not reported. | Employees |
| Satinsky et al., 2020 Maryland, US | Qualitative intervention study | Problematic substance use | Support linkage to community-based treatment, provide mentorship, support service navigation, promote retention in care and reduce barriers to engagement. | Peer delivered behavioural activation (BA) can be adapted with peer-delivered case-management, care linkage, and accessible community-friendly activities. | Background of problem substance use and recovery. | Not reported | Employees |

(*Continued*)

**Table 1.** (Continued)

| Authors Location | Study Type | Condition | Interventions | Outcome reported | Peer worker characteristics | Training provided | Employment status |
|---|---|---|---|---|---|---|---|
| Schel et al., 2022 Netherlands | Qualitative design | Homelessness | One-on-one mentorship | Positive self-image, personal growth and engagement with services. | All had personal experiences of being homeless; five female, five male; average age was 42 years; peer workers were included in this study when they provided support to homeless people in one-on-one relationships. | All peer workers received higher or intermediate professional training in peer support work or had almost finished this training. | Various (4 organizations formally employed, 1 organization employed on a voluntary basis). |
| Shah et al., 2018 Kenya | Pilot study | HIV | The peer navigator (PN) completed an initial encounter data collection form, discussed HIV prevention, assessed HIV status, and, offered counselling and linkage to HIV testing services. | There was a high prevalence of HIV among street-connected youth (SCY) engaging with PNs, especially among female SCY; high numbers of HIV-positive SCY initiated treatment, linked to care, and were still in care as at the end of April 2017. | One male, one female; living with HIV; aged 18–24 years; had greater than 1 year of recent experience of being street-connected. | Five-day training on HIV/AIDS prevention, treatment and care, reproductive health, counselling, and documentation. Ongoing mentoring and supervision by the social worker. | Employees |
| Stewart et al., 2007 Edmonton, Alberta | Cross-sectional | Homeless youth | Peers as part of social support network by providing info, modelling, and encouragement. | Sig. decreased loneliness. Qual. Results show increased support and coping. | Previously homeless youths who had made the transition from homelessness. | Not reported | Volunteers |
| Stewart et al., 2009 Edmonton, Alberta | Pilot intervention study | Homeless youth | Support groups, optional one-on-one support, group recreational activities, and meals; provide information, encouragement, or advice and acting as role models | Enhanced health behaviours, improved mental well-being, decreased loneliness, expanded social network, increased coping skills, enhanced self-efficacy, and diminished use of drugs and alcohol. | Youth (<18 years of age); past experience of homelessness | Trained by professional mentors (i.e., social workers, psychologists, and therapists) | Not reported |
| Surey et al., 2021 London, UK | Case study | Hepatitis C in people experiencing homelessness who inject drugs. | The HepCare project upskilled experienced peer support workers (PSWs) to become equal members of a service provider team, taking on advanced clinical roles normally carried out by medical/nursing specialists. | The HepCare model of inclusive working, promoting PSWs into an advanced role that utilizes their ability to negotiate the street interface and improves access to care. | Lived experience of transition from 'street' to 'institution'; all male, aged between 32 and 60, with between two and ten years of peer advocacy experience; all had experienced homelessness in their recent history, accompanied by substance misuse and mental health challenges. | Training in active case finding, testing, assessment, and referrals. New workers shadowed the nurse in the clinic doing the screening and tests. | Employees |

*(Continued)*

**Table 1.** (Continued)

| Authors Location | Study Type | Condition | Interventions | Outcome reported | Peer worker characteristics | Training provided | Employment status |
|---|---|---|---|---|---|---|---|
| Tseris, 2020 Sydney, Australia | Qualitative research study | Homelessness | Peer advisors were employed to work alongside caseworkers in a "shadowing" role in order to enhance the agency's capacity to provide support to service users | Peer advisers in health agencies add significant benefits to an agency's capacity to respond to the needs of service users. | People employed within agencies to use their personal experiences to both engage with service users and inform the work of staff whose practices are not explicitly informed by personal experiences. | Not reported | Employees |
| Weissman et al., 2005 Brooklyn, NY | Longitudinal | Homeless veterans with mental illness | Providing support, locating services, acting as mentors, and encouraging socialization with other participants in the program. | Participants with peer mentors were more likely to follow-up in treatment and increased socialization. | Formerly homeless, veterans, have severe mental illness; graduated from a homeless veterans' treatment program; have maintained sobriety in independent, stable housing for at least 1 year. | Training in relevant issues. After the initial training, PAs received weekly individual supervision with a licensed social worker. | Employees |

Peer and lay personnel were referred to using varied language including "peer-led", "peer-delivered," "community health worker," "peer support worker," "peer-driven interventions," "peer recovery coach," "peer-assisted case management," and "peer navigators." The characteristics of the peer worker varied based on the specific program. Almost all peer workers had a lived experience of homelessness, and experience with the condition or population of interest (i.e., type 2 diabetes, mental illness, addiction/substance use, hepatitis C, opioid use, HIV+ status, intimate partner violence, veterans, etc.).

Most programs [23] engaged peer health workers as employees [16, 18, 19, 22, 28, 30, 32, 34–39, 41–43, 46, 50, 52, 53, 55]. Four programs recruited the peer health workers as volunteers [14, 27, 45, 47]. One qualitative report noted in their study that four organizations formally employed peer workers, whereas one organization engaged peer workers on a voluntary basis [54]. Ten programs did not report employment status [14, 29, 31, 40, 46, 48, 49, 51, 56, 57]. 13 programs trained the peer workers via classroom and following a specified curriculum dependent on the programs' intervention [14, 19, 28, 30–32, 40, 43, 44, 49, 50, 52, 53]. For example, Resnik *et al.* reported that peer mentors underwent extensive training via a face-to-face 2-day meeting before starting subject recruitment, which included case management and peer mentor team interventions for homeless veterans [40]. Nine programs were run by an interdisciplinary team and the peer workers were supervised by registered health professionals including social workers, registered nurses, clinical nurse specialists, and clinical psychologists [18, 20, 29, 37, 39, 42, 46, 51, 53]. The remainder of the programs did not specify an approach to training [14, 16, 22, 27, 33–36, 38, 41, 45, 47, 48, 55–57].

## Interventions

Each study incorporated some form of peer support in the intervention. Rosen *et al.* reported on a mobile vaccination program to promote vaccine uptake using staff with lived experience of homelessness [19]. This resulted in increased COVID-19 vaccination uptake for PEH, as data collected through rapid field studies demonstrated that 16% of participants cited their

conversation with staff/peer workers as a primary reason for deciding to be vaccinated [19]. Davis *et al.* implemented peer-led diabetes education in a homeless community. They utilized field notes and post-implementation focus groups to determine that the intervention resulted in increased empowerment and knowledge of signs, symptoms, complications of diabetes, and diabetes medications [29]. Croft *et al.* implemented peer-delivered permanent supportive housing for individuals who were experiencing homelessness and issues with mental health/addiction [47]. In-depth semi-structured interventions were recorded, transcribed and analyzed using a grounded theory approach to determine that the intervention resulted in a decrease in psychological distress (i.e., decrease in number of emergency room psychiatric visits) [47]. Four studies implemented a peer-driven intervention for PEH who inject drugs and are HIV positive or hepatitis C positive. These resulted in increases in testing and medication adherence [38, 43, 48, 51]. Five studies utilized peer mentors who implemented support groups, one-on-one support, group recreational activities and provided encouragement/advice to homeless youth [22, 33, 44–46]. This supported participant engagement, enhanced health behaviours, improved mental well-being, decreased loneliness, expanded social networks, increased coping skills, enhanced self-efficacy, and diminished use of alcohol and drugs [22, 33, 44–46]. Moledina *et al.* reported that their peer support programs demonstrated no impact on housing relative to usual care; and no economic evidence was found for peer support [36]. Moledina *et al.* suggests that peer support cannot serve as a stand-alone intervention to promote stability [36].

### Enablers and inhibitors of peer and lay health worker programs for PEH

These programs reported four themes of enablers (shared experiences, trust and rapport, strong knowledge base, and flexibility of role) and five themes of barriers and inhibitors (lack of support and clear scope of role, poor attendance, precarious work and high turnover, safety, and mental well-being and relational boundaries). See Table 2 for a summary of enablers and inhibitors of peer and lay health worker programs among PEH.

### Enablers

**Shared experiences (n = 20).** The included studies show that due to their shared experiences, the peer support worker and program participants were able to have mutual understanding of one another and display empathy [14]. Shared experience appears to help clients to disclose their needs, and positions peers as facilitators [20, 50]. Due to their shared experiences, peers view clients as equals and reduce hierarchy in the process of accessing and receiving care [14]. In many studies, it was reported that peer support workers were able to motivate and engage clients who were previously unengaged in the program/treatment [32, 55]. In certain settings such as substance use and addiction, seeing recovery work for the peer made the possibility of recovery more tangible and achievable, and increased motivation for the participant's process of recovery [39]. Additionally, a non-judgmental approach and the foundation of trust helped to foster a safer emotional and physical space for interventions to take place [56]. Specifically, the relatability and decreased perceptions of stigma aided in this process [39]. For example, Salem *et al.* discussed how participants felt their disease (tuberculosis) was stigmatized, which influenced how they accepted treatment, and that peer workers helped to mediate and reduce that experience of stigma [39].

**Trust and rapport (n = 21).** A common theme in the included studies was the value of trust and reliability of the peer support worker, both with patients or clients and with other members of the health care or social support team [36]. This positions the peer worker to function as a bridge between people experiencing homelessness and program staff, and facilitate

**Table 2. Summary of enablers and inhibitors of peer and lay health worker programs among PEH.**

| Authors | Enablers | Inhibitors |
|---|---|---|
| Barker & Maguire, 2017 | Shared experience, role modelling, providing social support and increasing attendance. | Not stated |
| Barker et al., 2017 | Ability to connect based on shared experiences; development of trust. Viewing clients as equals; positive attitude/outlook. Program benefits for the peer support worker (i.e., increased self-esteem, confidence, and self-efficacy; gaining employment references, skill development, and other work possibilities). | Having to cope with challenging client behaviour. Exposure to potential triggers and risk of relapse Difficulty maintaining professional boundaries, while still being supportive. |
| Barker et al., 2018 | Developing trust based on shared experiences/experiential knowledge; viewing clients as equals. Reducing stigma, creating empowerment, role modelling, positive attitude, etc. Peers' ability to facilitate connections to other services and help. | Peers breaking boundaries (i.e., by being the clients' friend). Power imbalances between peers and clients. Peers overextending themselves and burning out. Exposure to potentially stressful situations and triggers. |
| Blonigen et al., 2022 | Effective communication and consistent messaging. Connecting patients to resources and helping navigate community-based care systems; more time dedicated to patients. The peer role itself as a facilitator (i.e., patients can better relate to peers and they identify with peers). | Other staff having lack of knowledge about the scope of the peer role; lack of communication between staff and peers. Length and intensity of intervention was viewed as insufficient. |
| Cerna et al., 2023 | Trust and relatable experience; sharing of previous life experience. Positive role modelling; empowering support; non-judgmental approach, openness, trust-building, listening and informality. Peers having personal preparedness such as previous work experience and/or training, a degree of recovery, and being open-minded. Peer support helps the PSW in their own recovery. | Staff retention Fulfilment of formal job requirements could be difficult for peers to navigate. Inability to maintain personal boundaries with clients. Non-peer staff members being unfamiliar about the PSWs approach and their roles. |
| Choi et al., 2022 | Peer ambassadors (PAs) address issues of access and mistrust through preexisting rapport and trust; knew the best language to use to discuss vaccines; were respected by their neighbours. Strong interpersonal skills (calm, kind, non-judgmental and confident). Community health workers (CHWs) saw potential for longer-term employment of program participants as PAs or even eventually as CHWs; connecting PAs with job opportunities and skills trainings. | Safety concerns Workload of CHWs Communication difficulties (i.e., unable to contact a peer for various reasons). Program retention Peer advisor not having access to transportation to work at different locations. |
| Corrigan et al., 2015 | Personal/shared experience; communication skills Having been homeless brings tolerance, dedication, passion and motivation to the role. | Navigators need to learn of resources available to peers to address health challenges and come up with practical approaches to access these resources; accessibility to resources |
| Crisanti et al., 2017 | Non-judgmental attitudes of peers. Ability to extend support by utilizing various resources (i.e., mental health, substance use, medical intervention and treatment resources). Strong interpersonal and functional skills. | Heavy caseload, lack of support from outside services, and long wait lists for programs. |
| Croft et al., 2013 | Peers as a patient advocate; supporting socially excluded patients through sharing their personal experience; gain trust. Cost-effective resource. Peers have a unique insight into the social dimensions of tuberculosis. | Not reported. |
| David et al., 2015 | Ability to motivate and engage clients who were previously unengaged in treatment. Ability to build trust with their clients at each client's pace, able to relate to shared experiences, and reducing stigma of diagnoses. | Tension among peer mentors and clients who had shared "street" histories including conflicts over romantic relationships. Barriers to treatment such as transportation and shelter. |
| Davis, et al., 2016 | Strong attendance and participation. Good relationship building skills and ability to provide a sense of empowerment. Peers having a strong knowledge base of diabetes information. | The complexity of the educational material. Language and speech barriers between peers and clients. |
| Deering et al., 2009 | Small group size allowed for increased cohesiveness and closer bond among members. | Large groups got split up, inhibiting the continuity and closeness and trust that women developed. |
| Erangey et al., 2020a | Mutual understanding and empathy, reducing stigma, increasing hope for recovery and creating a supportive environment. | Previous trauma, no desire/not ready to engage with program or change and non-supportive environments. |
| Erangey et al., 2020b | Peers being a familiar face, available, trustworthy and having a nonjudgmental attitude. Peers having shared experiences and ability to model recovery. Mutual aid between peers and clients. Flexibility of work and the peer role. | Dismissed by other service providers; power imbalances. Hard for others to identify them as peer support specialist Lack of attendance from youth. Stigma or objectification. Having to clarify the peer role to clients who want a friendship. |

(*Continued*)

**Table 2.** (Continued)

| Authors | Enablers | Inhibitors |
|---|---|---|
| Flike et al., 2020 | Program addresses the social determinants of health.<br>Program improves access to health care resources. | Not reported. |
| Fors & Jarvis., 1995 | Positive learning environment.<br>Preparedness and confidence of group leader.<br>Behavioural intention of clients to change. | Disruptive behaviour of clients. |
| Herts et al., 2020 | Ability of peers to form therapeutic relationships with their clients.<br>Able to foster trust and companionship.<br>Flexibility and accessibility. | Lack of adequate supervision and support.<br>High turnover among supervisors<br>Lack of affordable housing options and supports available for clients. |
| Kidd et al., 2019 | Peers' ability to be versatile and create a comfortable environment.<br>Peers' advocacy for clients.<br>Peers being good listener and their proximity to the youth in age and life experience. | Role confusion (i.e., participants struggled with defining the role of the peers initially).<br>Informal drop-in structures of engagement were difficult to plan, deliver, and were poorly attended. |
| MacLellan et al., 2017 | The building of rapport and self-disclosure of shared experiences.<br>Use of respect, reciprocity and friendship; good communication skills; showing acceptance of who the client is; mutual respect and the desire to work together.<br>When HCP staff understand the role of the PA, they work well together because the PA saves them time.<br>Being professional and non-judgmental. | PAs are often still vulnerable to the context within which they are now working (i.e., one PA could not work with clients in certain parts of the city due to his own recent challenging experiences there).<br>Potential exposure to triggers.<br>Lack of boundaries between PA and clients. |
| Magwood et al., 2019 | Ability of peers to provide physical and emotional safety (i.e., women's only programs).<br>A sense of empowerment, continuity and trust. | Patronizing attitudes and lack of commitment from workers.<br>Loss of autonomy. |
| Miler et al., 2020 | Leadership skill.<br>Creating a sense of community and connection.<br>Peer interventions can also benefit the peer workers themselves and lead to changes in their own behaviour. Able to build a special type of rapport and trust based on shared experience and lack of judgment. | Risk that peer support for people with history of substance abuse could lead to relapse.<br>The need to establish boundaries between peers and clients.<br>Problems with stigma and discrimination.<br>Peers feeling undervalued in their work; lack of clarity regarding the peer role; limited opportunities for career growth. |
| Moledina et al., 2021 | Ability to establish relationships based on trust due to their shared experiences. | Potential harms to peer workers: depression and anxiety symptoms. |
| Morris et al., 2020 | Reduced barriers to testing and treatment.<br>Effective communication skills.<br>Collaboration with the multidisciplinary team. | Not reported. |
| Nyamathi et al., 2021a | Able to provide a comprehensive continuum of care approach.<br>Caring peers; ability to offer compassion, support and dedication. | Not reported. |
| Nyamathi et al., 2021b | Ability to provide emotional support; ensuring confidentiality and trust.<br>Ability to share similar lived experiences.<br>Being an advocate for clients (being a spokesperson, translator, or accompanying PEH where they need to go). | Difficulty locating clients.<br>Stigmatization.<br>Food insecurity and lack of transportation jeopardize treatment initiation and completion. |
| Parkes et al., 2022 | The shared lived experience aided in enabling trusting, authentic and meaningful relationships to be developed.<br>The flexible role enabled peer navigators to work beyond the service they were based in. | Flexibility of roles sometimes contributed to tensions between existing staff and the peer navigators. |
| Ponce et al., 2014 | Creating safe emotional and physical spaces; non-judgemental approach.<br>Having peer workers trained in understanding and addressing trauma.<br>Accessibility—offering multiple meeting sites. | Participants unwilling or unable to engage in services for reasons related to their experiences with their violent/predatory partners.<br>Women reported that their partners did not want them to interact with male staff; fear of judgement; participants having a violent, coercive, or predatory partner. |
| Resnik et al., 2017 | Ability to relate to similar life experiences and build trust.<br>Ability to facilitate connection to services (housing, free telephone, clothing and bus tokens).<br>Good training provided to peers prior to start of program. | Peers' lack of knowledge (i.e., some Veterans thought that they understood less than they did about how to navigate services).<br>Clients not being open or receptive to services. |
| Rosen et al., 2023 | Ability to share personal stories and relate to mutual life experiences.<br>Having peers provide harm reduction supplies, food and hygiene kits. | Frequent encampment sweeps fostered government mistrust and made clients resist peer engagement and vaccine delivery. |

(*Continued*)

**Table 2.** (Continued)

| Authors | Enablers | Inhibitors |
|---|---|---|
| Salem at al., 2020 | Treatment readiness of clients. Ability of peers to provide Tuberculosis health education; keeping in contact with participants, building trust, keeping them engaged; and, familiarity with homeless population. | Lack of treatment readiness of clients; concurrent substance use; stigma. Negative perspectives on healthcare providers and institutions. s Inability to access population. |
| Satinsky et al., 2020 | Relatability and decreased perceptions of stigma. Seeing recovery work for the peer increases motivation to change. Ability for worker to link client to medication-based treatment models. | Lack of stability in clients' lives (i.e., housing instability/homelessness). Severity of clients' substance use. |
| Schel et al., 2022 | Increasing self-worth, self-esteem and confidence. Being accessible and available to clients. Having an equal relationship, role modelling (i.e., sharing own experiences to help clients) and ability to build a trusting rapport. | Role and time boundaries. |
| Shah et al., 2018 | Building rapport and trust with clients. Having a pre-existing relationship with the street connected youth in the community. Ability to make clients feel comfortable. | Clients not engaging in treatment. Instability of population; difficulty following up on test results as most did not have a mobile phone. Issues with stigma and discrimination. |
| Stewart et al., 2007 | Built relationships based on similar life experiences; encouragement and understanding. | Accessibility concerns such as lack of transportation, and program location. Lack of volunteer insurance for peer workers. |
| Stewart et al., 2009 | Strong coping and social skills; positivity. Ability to build trust with clients. | Psychological challenges such as depression, anxiety and negative emotions. Irregular attendance. |
| Surey et al., 2021 | Flexibility of peer work can present an opportunity to re-enter the labour market for a group that may struggle with fixed employment. Acceptance of efforts to include the peer support workers as part of the team (i.e., inclusive approach). Non-judgmental communication and trust gained by sharing their lived experience; ability to remove barriers to care. | Volunteer role is characterized by instability and insecurity. Barriers of power in the 'peer' relationship. Risk of relapse for the peer. As peers take on more advanced roles previously carried out by other health professionals, this can lead to conflict if not clearly defined. |
| Tseris, 2020 | Ability to develop nonhierarchical relationships with service users based on mutuality and shared experiences. Peer advisors gain benefits themselves such as gaining permanent employment. Ability to response to crises and to engage in risk assessment processes. | Precarious work opportunities/lack of job security, low pay, poor job-progression opportunities, and inadequate sick leave. Discriminatory attitudes from other staff members; stigma or positioned as less credible than "professional" viewpoints. Need for role clarification/ being tasked with work duties unrelated to the peer role. |
| Weissman et al., 2005 | Confidence of peer worker. Skill of gaining participants' trust. Ability to communicate with participants based on their having had similar life experiences. | High turnover of peer workers and lack of support from other professional staff. |

successful program design and successful client engagement with the program [48]. Peers were often trusted by clients as they knew the best language to discuss various health topics, and they were respected by the clients due to their shared experience/similar background [16].

**Strong knowledge base (n = 11).** The ability to reduce barriers to care and being knowledgeable of the disease/condition (i.e., diabetes, tuberculosis, etc.) was an important enabler of effective lay and peer worker programs [29, 39]. Peers acted as an advocate for clients to help address other factors that were impacting their health, as they had specific experiential knowledge of the condition that other providers did not [47]. For example, peer support workers were able to extend support by utilizing various resources such as mental health, substance use, housing support, medical interventions, and treatment resources [28, 40, 41]. Specifically, they were able to improve linkage to treatment by offering an acceptable, destigmatizing and flexible approach which addressed common barriers faced by low-income, minority individuals [41].

**Flexibility of role (n = 5).** Flexibility of the role enabled peers to work beyond the service where they were based [52]. There are few restrictions to how peers spend their time with

clients, therefore, they can tailor their approach specific to each individual client [35]. For example, peer support workers do not have to work from a specific location–they often engage with clients at locations of their choosing to increase accessibility to the program [35]. Peers were able to provide support to individuals at any given time based on their needs. This highlighted the importance of flexible and person-centered support for individuals that are severely and multiply disadvantaged [52]. Specifically, this follows the PIE approach, which refers to psychologically informed environments [52]. This involves developing psychological awareness of individual's needs; valuing training/support for staff; creating effective/safe service environments; and, focuses on the roles and responsiveness of the services to focus on improving relationships [52]. An additional benefit of role flexibility is that it allows peers to develop various skills which may present them the opportunity to re-enter the workforce for those who previously struggled with fixed employment [18].

## Inhibitors/barriers

**Lack of support and clear scope of role (n = 9).** A recurrent inhibitor identified was the heavy caseload, lack of support from outside services and long wait lists for the programs [28]. With the limited number of peer workers engaged in most programs, it was difficult for them to fully engage with the high volume of participants interested in the programs. The multidisciplinary team occasionally found it difficult to identify the peer support workers as a member of the care team. As a result, peer workers reported being treated in a dismissive manner by other service providers [33]. Role confusion was cited as an inhibitor as service providers and participants struggled with defining the role of the peers [44]. For example, the specific roles of peers in the early months of the intervention were less clear than was the case for other health care providers [44]. Role confusion contributed to tensions and conflicts between existing staff and the peer support workers [52, 55].

**Poor attendance (n = 7).** Lack of consistent engagement from peer and lay workers and irregular attendance from program participants (PEH) was identified as an inhibitor to program success [42, 46, 56]. Continuity of program staff was imperative to maintaining rapport and trust among the participants. On the other hand, program participants (PEH) did not have regular attendance for various reasons (i.e., unstable living conditions, transportation barriers, substance use, lack of readiness to fully engage in program, etc.), which hindered positive program outcomes [39, 41, 45, 46, 53].

**Precarious work and high turnover (n = 7).** Several reports noted high turnover of peer workers [28, 35, 42]. Reasons for high turnover may be due to precarious work opportunities, lack of job security, low pay, heavy workload, feeling undervalued, poor job-progression opportunities and inadequate sick leave [16, 18, 55, 57]. Many peer support positions are temporary, contract-based, or grant-funded, which creates a constant sense of job insecurity. Additionally, the heavy workload may lead peer workers to feel overwhelmed and undervalued, leading to burnout and a desire to leave their positions.

**Safety (n = 4).** Safety of peer support workers while actively engaged in the role was cited as a common inhibitor. Various programs utilized incentives for treatment and testing, and peer support workers would be responsible for carrying gift cards, cash, etc. which positioned them as targets for theft [16]. Some peer support workers had prior conflicts with the community they were serving, such as violent encounters and conflict over intimate relationships, which introduced safety concerns while performing job duties [16, 32].

**Mental well-being and relational boundaries (n = 9).** Various studies reported peers had difficulty maintaining professional boundaries with clients, while still being supportive [14, 48, 50, 54, 57]. Challenges with personal boundaries would occasionally occur where peers

developed friendships, lent money, borrowed items, etc. [49, 50]. To maintain a therapeutic relationship, peers would face the delicate task of clarify their role to clients without eliciting shame or rejection, especially when clients were seeking a friendship or more personal relationship [33]. On the other hand, peers would clarify that their roles were also not the same as professional clinicians either, and that professional approaches to boundaries could not be transposed into peer or lay work, especially as they shared similar lived experiences as their clients. Peers who had a history of substance abuse and were working to support clients with substance use concerns reported a risk of relapse [18, 57]. Some studies reported that peer work had the potential to negatively impact peers' mental health, including worsening symptoms of depression and anxiety [36, 46].

## Discussion

This scoping review identified 38 sources of evidence describing enablers and inhibitors to peer and lay health work programs in the homeless sector published between 1995 and 2023. The majority of the studies were published after 2017, which indicates new research is rapidly emerging on this topic.

The included programs were effective when they were able to build trusting relationships with staff and clients and reduce the barriers that PEH faced when accessing programs. Peers having an intimate knowledge of homelessness and shared lived experiences regarding the condition/disease of interest, led to the ability to connect and build strong rapport [14]. In terms of programs related to substance use/recovery, peers had mutual understanding and empathy, were able to reduce stigma, and increase hope for recovery by creating supportive environments for clients [22]. Rather than clients having to travel to program locations, peers were able to improve accessibility by meeting clients at locations they chose (i.e., the homeless shelter, coffee shops, parks, etc.) [35].

A critical element of effective peer and lay health support programs is the concept of mutual benefit: both peers and clients benefit from utilizing these programs. Peer support work programs provide mutual benefit to both peer workers and clients because they create a sense of empathy and understanding between them. By working together, they have the capacity to form supportive communities that help clients build resilience, reduce isolation, and improve their overall mental and physical health. Additionally, clients benefit from the expertise and knowledge from peer workers, who provide them with practical strategies to manage their health and help them navigate the healthcare system. Peer support work programs provide a sense of purpose and fulfillment for peer workers, who can use their own experiences to help others while gaining valuable training and skill development. Throughout this process, peer support workers also gain employment references and opportunities for other work opportunities which in turn, may help bring them out of homelessness [14].

Peer programs create a safe and inclusive emotional and physical environment where participants can talk freely about their shared lived experiences. Peer support work programs are beneficial to individuals experiencing homelessness in numerous ways such as providing emotional support, encouragement, and motivation. These programs also offer practical assistance, such as food, shelter, healthcare services, and housing and employment assistance to provide support to PEH. Peer support work programs empower individuals to take responsibility for their own lives by building resilience, confidence, and taking control of their health. They provide a platform where individuals can share their experiences and success stories, which can significantly contribute to the overall process of ending homelessness through a multidisciplinary and collaborative approach.

## Limitations

This scoping review has limitations. First, while two reviewers independently undertook the screening and document selection process, only the lead author undertook the data extraction and charting process. However, we do not believe that this compromises the rigor of this review due to the nature of the evidence selected. Secondly, the quality of included documents/studies were not assessed, therefore, recommendations cannot be given based on the quality of the selected evidence. A final limitation of this scoping review was restricting the search to include only articles written in English, as the inclusion of other languages may have potentially resulted in useful information and perspectives on the topic.

## Conclusion

This review demonstrates that peer and lay health programs have been successfully implemented to serve people experiencing homelessness. Overall, peer and lay health work programs are effective for people experiencing homelessness in various contexts (i.e., substance use, chronic disease management, harm reduction, mental health, etc.). These programs may experience challenges due to the many barriers that PEH experience when trying to access and maintain participation in services. Many programs were also able to report facilitating factors that were able to overcome these inhibitors and implement successful peer and lay support programs. Further investigation is needed to understand how peer and lay support work programs are successfully implemented within the homeless population in different contexts (i.e., substance use, transitional housing, mental health, etc.) and to build best practice recommendations for these programs. Organizations seeking to implement these interventions should anticipate and plan around known facilitators and barriers to promote success of the program.

## Supporting information

**S1 Checklist. PRISMA checklist.**
(DOCX)

**S1 Text. MEDLINE search strategy.**
(DOCX)

## Author Contributions

**Conceptualization:** Jessica Mangan, Amna Siddiqui.

**Data curation:** Jessica Mangan, Pablo del Cid Nunez, Amna Siddiqui.

**Formal analysis:** Jessica Mangan.

**Investigation:** Jessica Mangan, Pablo del Cid Nunez, Amna Siddiqui.

**Methodology:** Jessica Mangan, Pablo del Cid Nunez, Sara Daou, Amna Siddiqui, Aaron M. Orkin.

**Supervision:** Graziella El-Khechen Richandi, Aaron M. Orkin.

**Writing – original draft:** Jessica Mangan, Sara Daou, Amna Siddiqui.

**Writing – review & editing:** Jessica Mangan, Pablo del Cid Nunez, Sara Daou, Graziella El-Khechen Richandi, Amna Siddiqui, Jonathan Wong, Liz Birk-Urovitz, Andrew Bond, Aaron M. Orkin.

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
