## [Decision Letter · Decision Letter 0]

19 Feb 2024

PGPH-D-24-00025

Peer and lay health work for people experiencing homelessness: A scoping review

Dear Dr. Mangan,

Thank you for submitting your manuscript to PLOS Global Public Health. After careful consideration, we feel that it has merit but does not fully meet PLOS Global Public Health’s publication criteria as it currently stands. Therefore, we invite you to submit a revised version of the manuscript that addresses the points raised during the review process.

Please see the comments from two reviewers below. Both reviewers appear very positive about the contribution of the work, and have provided suggestions for how to strengthen the manuscript.

We look forward to receiving your revised manuscript.

Kind regards,

Hanna Landenmark

Staff Editor

Journal Requirements:

1. We noticed you have some minor occurrence of overlapping text with the following previous publication(s), which needs to be addressed:

- DOI: 10.1002/cl2.1154

-https://www.pathhomekingston.ca/homelessness-in-our-community/

-Stewart, Miriam & Reutter, Linda & Letourneau, Nicole & Makwarimba, Edward. (2009). A Support Intervention to Promote Health and Coping Among Homeless Youths. The Canadian journal of nursing research = Revue canadienne de recherche en sciences infirmières. 41. 55-77. 

In your revision ensure you cite all your sources (including your own works), and quote or rephrase any duplicated text outside the methods section. Further consideration is dependent on these concerns being addressed.

Additional Editor Comments (if provided):

Reviewers' comments:

Reviewer's Responses to Questions

**Comments to the Author**

1. Does this manuscript meet PLOS Global Public Health’s publication criteria? Is the manuscript technically sound, and do the data support the conclusions? The manuscript must describe methodologically and ethically rigorous research with conclusions that are appropriately drawn based on the data presented.

Reviewer #1: Yes

Reviewer #2: Yes

2. Has the statistical analysis been performed appropriately and rigorously?

Reviewer #1: N/A

Reviewer #2: N/A

3. Have the authors made all data underlying the findings in their manuscript fully available (please refer to the Data Availability Statement at the start of the manuscript PDF file)?

Reviewer #1: Yes

Reviewer #2: Yes

4. Is the manuscript presented in an intelligible fashion and written in standard English?

Reviewer #1: Yes

Reviewer #2: Yes

5. Review Comments to the Author

Reviewer #1: Dear Authors,

Thank you for the opportunity to review your submission. I really enjoyed reading it. I have provided the following comments in the spirit of strengthening your paper so it can maximise it's contribution to this important field.

1. Title - could the title be more informative? At the moment it tells the reader that it's about peer workers, homelessness and it's a scoping review. I wonder whether including the outcome, by saying, Impact of peer worker programs on service access for people experiencing homelessness?

2. Definition of peer and lay workers - I recommend removing 'are individuals without professional training' because they may have education in other fields, just not homelessness or social work.

3. Question/objectives - could this be slightly more clear by using the PiCO approach - population, intervention, context, outcome. The aim of the review is a little busy, perhaps the aim could be to explore the impact of lay and peer worker programs....

The concept explored didnt quite match the aim, because the effectivenens of peer workers is stated here and it isnt in the aim. The outcome is about factors that hinder or facilitate, so perhaps this could all be captured in the aim.

4. Information sources - CINAHL not CINHAL. In developing the search strategy, who was it that peer reviewed it? Were these other authors, a research librarian or people with lived experience?

I recommend including the search terms in the paper itself.

5. Selection of sources - page 9 - the term 'characteristics of effective peer and lay programs...' can you be more specific and name who is deciding whether they are effective and on what basis? Is it effective in relation to housing or health access, or effective in terms of building rapport?

The paragraph on types of study included could perhaps be deleted because the first line of this section states that any study design was included.

Why did you dedulicate prior to uploading to COVIDENCE, when COVIDENCE does this automatically?

In screening the papers could you provide more detail on the review process in relation to disagreements. Which authors undertook screening, which authors then reviewed disagreements and what was the definition of consensus - was it 75% of the authors agreed?

6. Data extraction - it is a limitation if only one author extracted data - did anyone check this for accuracy? Was the synthesis of the data also undertaken by one author? This is also a limitation.

7. Results - It seemed a bit unusual that literature reviews were included in the included papers of a scoping review. From experience this can make data extraction very complex, and sticking with primary research can be a more streamlined approach.

Table 1 - It would be helpful to provide more information regarding the column headings. I wouldn't consider homelessness as a condition, more a circumstance or experience. Homelessness was also an inclusion criteria so I would expect all papers to relate to homelessness, therefore no need for a column. Also, column heading outcome reported, this is not very detailed, what do you mean specifically by social outcomes, or health outcomes, were these improvements in certain conditions, or access to housing. This outcomes column is important as it tells use the potential impact of peer support workers. As a suggestion, perhaps include the title of each paper in the table because that will give the reader a lot more information, and it would also be helpful to understand the study aim too. The column about employment status is not particularly informative either, and perhaps this is something that could be removed from the table and discussed in the narrative instead. From reading the table I really dont get a sense of the papers included.

The descriptions of the themes of the studies is informative. There could be slightly more detail, for example, how many studies were from which countries, how many were cohort studies - simply place (n=x) after each statement.

8. Table 2 is informative. Consider adding some examples e.g., how was stigmatisation an inhibitor - who was experiencing the stigma or from whom was this behaviour perceived?

Rather than state 'Poor attendance' as a heading, consider simply stating 'Attendance' otherwise it might be perceived as slightly judgmental. There will be lots of genuine reasons why people dont attend, many of which will be out of the person's control.

9. Discussion. Please review the areas of text without references. For example, page 25, starting A critical element... the text really needs supporting with references. Likewise the subsequent paragraph. Consider adding a section in the discussion around the future of peer workers and given their impact, ways that we can employ them in the field of homelessness. Do we need policy change, education programs, government funding?

10. Limitations - only having one person extract data definitely impacts rigour - it is good this is acknowledged, I recognise deleting the statement about the compromise of the rigor. Is it a limitation that there were no RCT's?

Best wishes

Reviewer #2: Thank you for the opportunity to review this important piece of work. As someone who is currently working with a few difference homelessness organisations who are in their infancy in implementing peer workers within their models, this review is highly beneficial to summarising the work internationally in this field!

The background section appears to be missing something between “homelessness” paragraph and “scope”. A sentence or two summarising the literature on peer work in homelessness would be beneficial here to give better flow as currently it feels a bit jarring to go from describing challenges associated with homelessness to the enablers of a peer workforce.

Another term you could add in is “expert by experience” – colleagues of mine in the UK largely used this term.

Under questions and objectives you refer to “the participants in this review” – as they aren’t participating in anything – perhaps “population of interest” or similar would better describe this.

Under Selection of sources of evidence - some of the paragraphs could be simplified (or even deleted) by creating a table with a column for inclusion, and a column for exclusion criteria.

Last sentence of last para in selection of sources- regarding consensus of conflicts – was this consensus made between the two reviewers? Or was a third author brought in?

Characteristics table – Few inconsistencies to check, i.e., in the Sydney study they are referred to as advisors and advisers. Sometimes refer to the participants and “youth” and sometimes as “young people. Sometimes “not reported” is followed with a full-stop, but most of the time not. This may be one to check with the editor, but I find having a footnote at the end of the table with all the abbreviations in useful

Populations and personnel – “almost all peer workers had a lived exp of homelessness” – how many?

Table 2 – I’m hesitant to raise this one as I do find the information in the table very useful! But another way this information could be presented is as a tick box matrix table whereby all the studies are listed in the first column, then each of the types of barriers/enablers are listed in the first row. (i.e., column two might be “trust & relationships” and all studies that discuss that as an enable have a tick). This would make it a bit more user friendly if someone was looking for specific barriers/enablers within peer work in homelessness to find the articles of relevance to them as currently would need to look through the table to try and identify the 21 articles yourself as they aren’t specifically referenced in the paragraph on this. Anyway, please see this as a suggestion only rather than a requirement!

Discussion – “the majority of studies were published after 2017”… would be useful to have a number or % in brackets.

Discussion – some commentary around overall trends in homeless peer workers would be beneficial. For example, in Australia, while there is wide acceptance of peer workers in MH and AOD, homelessness is still very much in its infancy and there are few examples of where its working well. While there may be some small examples of services that have a peer worker (or two), I know there has been an overall feeling around the general lack of support/guidance, and feeling “of being on their own”, training programs haven’t really taken off here and organisations are resource poor to implement this – which of course can have grave consequences around feelings of re-traumatisation on sharing their experiences and crossing those professional boundaries etc.. Where we know of individuals doing this sort of work we have connected them so they can share experiences/resources etc., but this is all very small scale.

Looking forward to reading the final piece.

6. PLOS authors have the option to publish the peer review history of their article (what does this mean?). If published, this will include your full peer review and any attached files.

**Do you want your identity to be public for this peer review?** For information about this choice, including consent withdrawal, please see our Privacy Policy.

Reviewer #1: No

Reviewer #2: **Yes: **shannen vallesi

While revising your submission, please upload your figure files to the Preflight Analysis and Conversion Engine (PACE) digital diagnostic tool, https://pacev2.apexcov

---

## [Editor Report · Decision Letter 1]

21 May 2024

Enablers and inhibitors of effective peer and lay health work for people experiencing homelessness: A scoping review

PGPH-D-24-00025R1

Dear MS Jessica Mangan

We are pleased to inform you that your manuscript 'Enablers and inhibitors of effective peer and lay health work for people experiencing homelessness: A scoping review' has been provisionally accepted for publication in PLOS Global Public Health.

Best regards,

Priyamvada Paudyal, PhD

Academic Editor